# Tactile DreamFusion:
# Exploiting Tactile Sensing for 3D Generation

**Ruihan Gao**[1]   **Kangle Deng**[1]   **Gengshan Yang**[1]   **Wenzhen Yuan**[2]   **Jun-Yan Zhu**[1]
[1]Carnegie Mellon University    [2]University of Illinois Urbana-Champaign
https://ruihangao.github.io/TactileDreamFusion/

## Abstract

3D generation methods have shown visually compelling results powered by diffusion image priors. However, they often fail to produce realistic geometric details, resulting in overly smooth surfaces or geometric details inaccurately baked in albedo maps. To address this, we introduce a new method that incorporates touch as an additional modality to improve the geometric details of generated 3D assets. We design a lightweight 3D texture field to synthesize visual and tactile textures, guided by 2D diffusion model priors on both visual and tactile domains. We condition the visual texture generation on high-resolution tactile normals and guide the patch-based tactile texture refinement with a customized TextureDreambooth. We further present a multi-part generation pipeline that enables us to synthesize different textures across various regions. To our knowledge, we are the first to leverage high-resolution tactile sensing to enhance geometric details for 3D generation tasks. We evaluate our method in both text-to-3D and image-to-3D settings. Our experiments demonstrate that our method provides customized and realistic fine geometric textures while maintaining accurate alignment between two modalities of vision and touch.

## 1   Introduction

Generating high-fidelity 3D assets is crucial for a wide range of applications, from content creation in gaming and VR/AR to developing realistic simulations for robotics. Recently, a surge in emerging methods has enabled the creation of 3D assets from a single image [1] or a short text prompt [2, 3]. Their success can be attributed to advances in generative models [4, 5] and neural rendering [6, 7], as well as the availability of extensive 2D and 3D datasets [8, 9].

Although existing methods can effectively capture the overall shape and visual appearance of objects, they often struggle with synthesizing fine-grained geometric details, such as stochastic patterns or bumps from the material, as shown on the left side of Figure 1. As a result, the final mesh output tends to be either overly smooth (e.g., the avocado in the top row) or has geometric details incorrectly baked into the albedo map (e.g., the beanie in the bottom row). But why is that?

We argue that there are two major bottlenecks. First, high-resolution geometric data are largely absent in current 2D and 3D datasets. For image datasets like LAION [8], obtaining geometric details is challenging due to the limited camera resolution in capturing real-world objects. In the case of 3D asset datasets like Objaverse [9], although the assets often include visual textures, they rarely feature high-resolution geometric textures. Second, it is difficult for humans to precisely describe fine geometric textures in natural language, making text-based applications even more challenging.

To address these issues, we propose exploring tactile sensing to capture high-resolution texture data for 3D generation. Given a text prompt or an input image, we first generate a base mesh with an albedo map as our 3D representation. We then capture detailed surface geometry of the target texture

38th Conference on Neural Information Processing Systems (NeurIPS 2024).

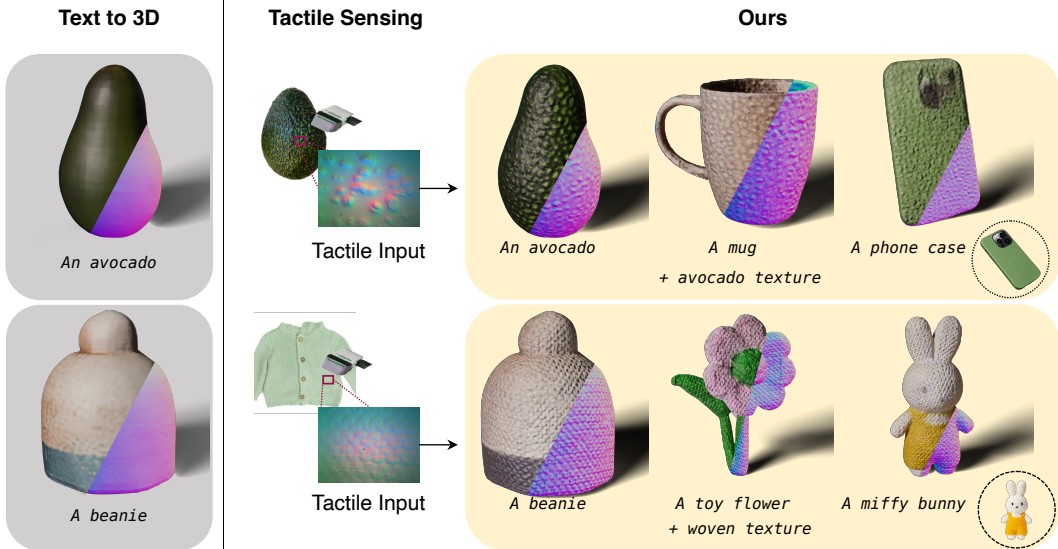

| Text to 3D | Tactile Sensing | Ours |
|---|---|---|

Figure 1: Our method leverages tactile sensing to improve existing 3D generation pipelines. *Left*: Given a text prompt, we first generate an image using SDXL [12] and then run Wonder3D [13] to generate mesh from the image. This process often results in a mesh with an overly smooth surface. *Right*: Our method takes a text prompt and several tactile patches and generates high-fidelity coherent *visual* and *tactile* textures that can be transferred to different meshes. Our method can easily adapt to image-to-3D tasks, as shown in the rightmost column, with the reference image's thumbnail displayed at the bottom right corner. Please visit our webpage for video results.

using GelSight [10, 11], a high-resolution tactile sensor. We then convert these tactile data into normal maps and train a TextureDreambooth on these normal maps. To refine the albedo map and ensure alignment between visual and tactile modalities, we learn a lightweight 3D texture field that co-optimizes the albedo and normal maps using 2D diffusion guidance across both domains.

Furthermore, we extend our approach to synthesize textures across multiple object regions by aggregating 2D diffusion-based part segmentation in a 3D label field.

Our experiments demonstrate that our method outperforms existing text-to-3D and image-to-3D methods in terms of qualitative comparison and user study. Our method ensures coherent high-resolution textures with precise alignment between appearance and geometry, as shown on the right side of Figure 1. Our code and datasets are available on our project website.

## 2  Related Work

**3D generation.**  Following the success of text-to-image models [14, 15, 16, 17], we have witnessed a booming development of 3D generative models conditioned on text or images. Recent works have used 2D diffusion priors in 3D generation [2, 18, 3, 19, 20, 21, 22, 13, 23], with notable methods like DreamFusion [2] introducing Score Distillation Sampling (SDS) to optimize a 3D representation using gradients from 2D diffusion models. Follow-up works have further extended SDS optimization using multi-view diffusion models, improving both 3D generation and single-view reconstruction [1, 24, 25, 26, 27, 28, 13, 29, 30].

Another line of research [31, 32, 33, 34] trains large-scale transformers to generate 3D shapes in a feed-forward manner, requiring high-quality large-scale 3D asset datasets. While these models effectively capture global shapes, they often fail to accurately render fine-grained geometric details, which are either incorrectly baked in color or lost entirely.

**Geometry representation in 3D generation.**  Researchers have explored various 3D representations for generation, such as voxel grids [35, 36, 37], point clouds [38, 39, 40], meshes [3, 41, 23, 42, 43, 44, 45, 19], neural radiance fields (NeRF) [46, 47, 2, 20, 21], neural surfaces [48, 22, 13, 25], and Gaussians [49, 50]. Among these, meshes support faster and more efficient rasterization rendering than volumetric rendering. Meshes also integrate seamlessly with graphics engines for downstream

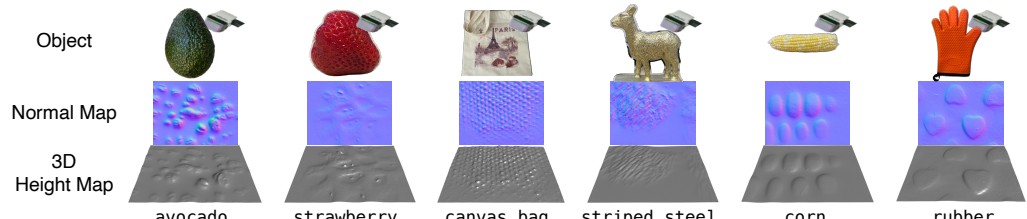

Figure 2: `TouchTexture` **dataset.** We collect tactile normal data from 18 daily objects featuring diverse tactile textures. To demonstrate the local geometric intricacies, we show the tactile normal map and a 3D height map for each object. Please refer to the supplement for the full set of our data.

applications. In contrast, volume representations require separate post-processing to extract meshes, often resulting in a loss of quality. Our approach uses meshes to represent coarse geometry while embedding local geometric details into a 3D texture field of tactile normal.

**3D texture generation and transfer.** Simultaneously generating geometry and texture often results in blurry textures due to surface smoothing or averaging across views. To address this, many approaches propose a subsequent stage that focuses on refining high-resolution textures [22, 13] or treats texture generation as a separate problem entirely [51, 52, 53, 54, 55, 56]. Additionally, various texture transfer methods enable the transfer of texture to new shapes using images [57, 58, 59, 60] or other material assets [61]. While most methods focus solely on visual appearance, our approach produces high-fidelity geometric details. Closely related to our method, NeRF-Texture [62] also transfers textures with geometric variations but requires 100-200 images of the same object. In addition, their method models the mesostructure with a signed distance and a coarse normal direction at each projected vertex. Hence, it can model sparse structures at a centimeter scale but not delicate patterns on the object's surface. In contrast, our method leverages high-resolution tactile sensing at the millimeter scale and provides accurate surface normal details, with richer details at fine scales.

**Tactile sensing in 3D.** Tactile data is useful in providing contact information and are widely used in 3D perception and robotics tasks. Early works start with reconstructing simple objects [63, 64, 65, 66], and then evolve to more complicated scenes with latest 3D neural representations [67, 68], as well as robotic in-hand reconstruction with multi-finger contacts [69, 70]. Vision-based tactile sensing [10, 71, 11], a particular sensing mechanism based on the photometric stereo principle, can provide high-resolution surface texture information like surface normals and thus can be useful for high-quality 3D synthesis and generation. Several recent works have also used it for 2D generation [72, 73, 74] and 3D scene reconstruction [75, 67, 76]. In contrast, to our knowledge, our work is the first to use tactile sensing for 3D generation.

## 3 Tactile Data Acquisition

We use GelSight [10, 11] to acquire high-resolution geometric details by touching the object's surface. The raw sensor data are then pre-processed to obtain high-frequency signals, as shown in Figure 3.

**GelSight tactile sensor.** The GelSight sensor is a vision-based tactile sensor that uses photometric stereo to measure geometry at a high spatial resolution at the contact surface. It can capture the fine details on the surface, such as bumps on avocados' skin and patterns of crochet yarns. In our work, we use the GelSight mini with a sensing area of 21mm $\times$ 25mm ($h \times w$) and a pixel resolution of $240 \times 320$, equivalent to about 85 micrometers per pixel. The sensor captures an RGB image, which can be mapped to the surface normals and then used to reconstruct a surface height map.

**Tactile data pre-processing and contact mask.** We manually press the GelSight sensor against the object's surface to obtain a tactile image over a small area. The derived height map contains both a low-frequency component, representing the surface's global shape, and high-frequency geometric textures. We first apply a high-pass filter to the height map to extract texture information and then center-crop the masked region to a square patch to remove any non-contact area. Finally, we convert the masked height map patch back to a normal map patch by computing its gradients.

Figure 2 shows our dataset `TouchTexture` collected from 18 daily objects with diverse tactile textures. Our dataset is available on our project website.

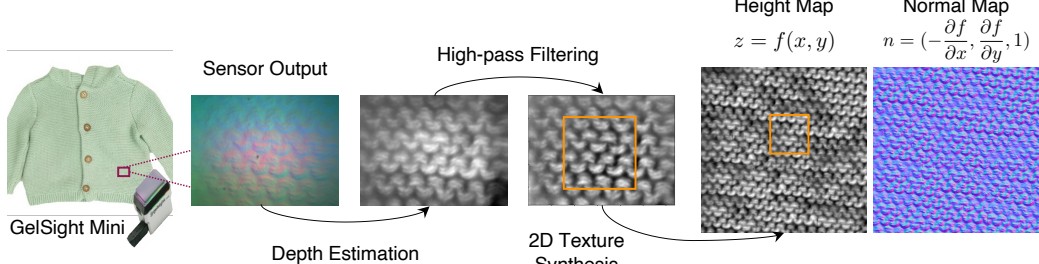

Figure 3: **Tactile data capture.** We collect one patch by pressing GelSight Mini on an object surface. We use Poisson integration to estimate the contact depth from the sensor output, apply high-pass filtering to extract the high-frequency texture information, and then run the 2D texture synthesis method of Image Quilting [77] to obtain an initial texture map. Finally, we convert the height map back to a normal map.

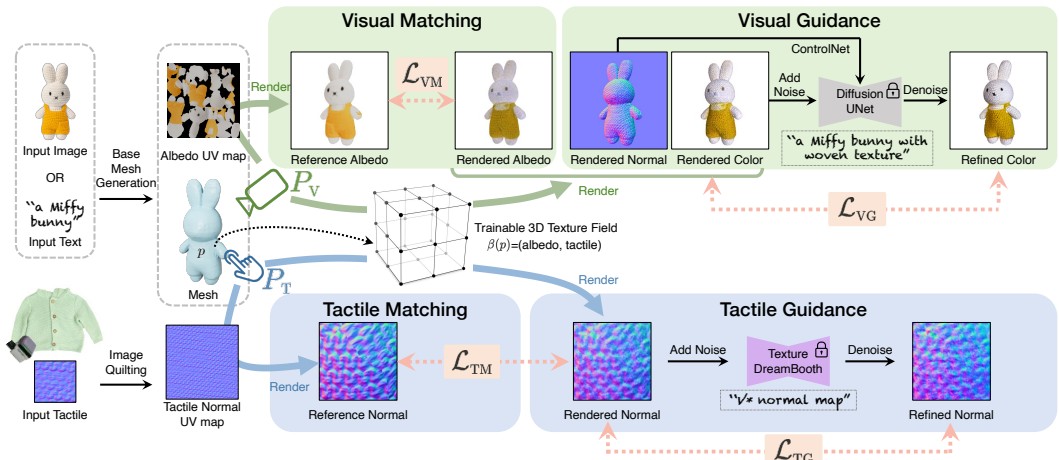

Figure 4: **Method overview.** Given an input image or a text prompt, our method generates a mesh with high-quality visual and normal texture. We first generate a base mesh with albedo texture using a text- or image-to-3D method. We use a 3D texture field with hash encoding to represent albedo and tactile normal textures and train it with loss functions on rendered images. To capture the scale differences between visual and tactile modalities, we sample distinct camera views, $P_V$ for visual rendering and $P_T$ for tactile rendering. For texture refinement, we train the texture field with a visual matching loss $\mathcal{L}_{VM}$, to ensure fidelity to the input mesh, and a visual guidance loss with normal-conditioned ControlNet, $\mathcal{L}_{VG}$, to enhance photorealism and cross-modal alignment. We further apply a tactile matching loss, $\mathcal{L}_{TM}$, and a tactile guidance loss, $\mathcal{L}_{TG}$, using a customized Texture Dreambooth, to achieve high-quality geometric details aligned with the distribution of tactile input $V*$ texture exemplars.

## 4 Method

Generating 3D assets with high-resolution geometric details is challenging due to the difficulty of acquiring low-level texture details from text prompts or a single image. Therefore, we incorporate an additional input modality, tactile normal maps, to enhance the geometric details in 3D asset generation. Figure 4 shows our overall pipeline. Our method takes an input image and a tactile normal patch as input and generates a 3D asset with a high-fidelity, color-aligned normal map. The input image can be a real image or generated by a text-to-image model such as SDXL [12]. Below, we first describe how we generate the base mesh in Section 4.1. We then introduce our core texture refinement algorithm in Section 4.2 and extend it to objects with multiple materials in Section 4.3.

### 4.1 Base Mesh Generation

Similar to prior works with stage-wise geometry sculpting and texture refinement [22, 49], we first generate a base mesh with albedo texture using a text- or image-to-3D method. We then unwrap a UV map of the exported mesh $M$ and project the vertex albedo onto an albedo UV map.

**A 2D texture transfer baseline.** To incorporate the tactile information, one could synthesize a large 2D texture map with an exemplar of the preprocessed normal patch using 2D texture synthesis algorithms such as image quilting [77] and use it as a normal UV map of the object mesh. However,

this would result in inconsistencies between visual and tactile textures, especially when transition in visual color indicates geometric change, as shown in Figure 9.

## 4.2 Texture Refinement

To address this issue, we jointly optimize the albedo and the normal tactile texture to ensure alignment.

**Texture representation.** For ease of optimization, we represent the visual and tactile normal textures as a 3D texture field. In practice, we use a multi-resolution hash grid [78] $\beta(\cdot)$, whose input is a 3D spatial coordinate $p$ on the mesh:

$$\beta(p) = (\mathbf{c}, \mathbf{n}_{\mathrm{T}}), \tag{1}$$

where $\mathbf{c} \in \mathbb{R}^3$ is the albedo and $\mathbf{n}_{\mathrm{T}} \in \mathbb{R}^3$ is the tactile normal defined in the tangent space on the mesh surface.

Given the base mesh $M$, the texture field $\beta(\cdot)$, a camera pose $P$, and a lighting condition $L$, we can render a color image $\hat{\mathbf{I}}_{\mathrm{C}} \in \mathbb{R}^{H \times W \times 3}$ of the object using a differentiable rasterizer $\mathcal{R}$, nvdiffrast [79]:

$$\hat{\mathbf{I}}_{\mathrm{C}}(P) = \mathcal{R}(M, \beta(\cdot), P, L), \tag{2}$$

where $H$ and $W$ represent image height and width, respectively.

Specifically, for each 3D point on the mesh, we composite the base normal $\mathbf{n}_{\mathrm{B}}$ from the mesh geometry and the tactile normal $\mathbf{n}_{\mathrm{T}}$ from the texture to get the shading normal $\mathbf{n}$:

$$\mathbf{n} = \mathbf{Q}_{\mathrm{TBN}} \cdot \mathbf{n}_{\mathrm{T}} = [\mathbf{t}, \mathbf{n}_{\mathrm{B}} \times \mathbf{t}, \mathbf{n}_{\mathrm{B}}] \cdot \mathbf{n}_{\mathrm{T}}, \tag{3}$$

where $\mathbf{n}_{\mathrm{B}}$ is the global geometric normal defined by the base mesh, $\mathbf{t}$ is the tangent vector, and $\mathbf{Q}_{\mathrm{TBN}}$ is the Tangent-Bitangent-Normal (TBN) Matrix for every surface point. Given the calculated shading normal and a sampled camera pose $P$, we use a point light and a simple diffuse shading model to produce the color image $\hat{\mathbf{I}}_{\mathrm{C}}(P)$ following prior works [2, 3]. We can also obtain $\hat{\mathbf{I}}_{\mathrm{A}}(P)$, $\hat{\mathbf{I}}_{\mathrm{T}}(P)$, and $\hat{\mathbf{I}}_{\mathrm{N}}(P)$ by projecting the albedo, tactile normal, and shading normal onto a sample view $P$.

**Learning 3D texture field.** Given the scale discrepancy between vision and touch, we sample camera views differently for two modalities. To supervise visual renderings $\hat{\mathbf{I}}_{\mathrm{C}}$ and $\hat{\mathbf{I}}_{\mathrm{A}}$, we sample camera poses $P_{\mathrm{V}}$ orbiting the base mesh looking at the object center with perspective projection. To supervise tactile renderings $\hat{\mathbf{I}}_{\mathrm{T}}$, we sample camera poses $P_{\mathrm{T}}$ close to mesh surfaces based on vertex normals to emulate the captured data using a real sensor. Specifically, we randomly sample a vertex, place a camera along this vertex's normal direction, and set the camera to look at the sampled vertex with orthographic projection.

We initialize the neural texture field of albedo with a reconstruction loss of rendered images:

$$\mathcal{L}_{\mathrm{VM}} = \left\| \hat{\mathbf{I}}_{\mathrm{A}}(P_{\mathrm{V}}) - \mathbf{I}_{\mathrm{A}}(P_{\mathrm{V}}) \right\|_2^2, \tag{4}$$

where $\mathbf{I}_{\mathrm{A}}$ is rendered using the exported albedo UV map in Section 4.1 from the same camera $P_{\mathrm{V}}$. Similarly, we initialize the tactile normal texture field with a **T**actile **M**atching (TM) loss:

$$\mathcal{L}_{\mathrm{TM}} = 1 - \cos\left( \hat{\mathbf{I}}_{\mathrm{T}}(P_{\mathrm{T}}), \mathbf{I}_{\mathrm{T}}(P_{\mathrm{T}}) \right), \tag{5}$$

where $\mathbf{I}_{\mathrm{T}}$ is rendered using the tactile normal UV map from the image quilting results from a tactile camera pose $P_{\mathrm{T}}$. Here, we use image quilting [77], a patch-based texture synthesis algorithm, to synthesize a reference texture map $\mathbf{I}_{\mathrm{T}}$ that roughly matches the physical scale of real materials and use it for initialization.

To refine the visual texture, we use a diffusion guidance loss inspired by the multi-step denoising process in SDEdit [80], Instruct-NeRF2NeRF [81], and DreamGaussian [49]. Different from existing works, we compute the **V**isual **G**uidance loss $\mathcal{L}_{\mathrm{VG}}$ based on a normal-conditioned ControlNet [82] to ensure the refined visual texture is consistent with the tactile normal. Given the rendered color image $\hat{\mathbf{I}}_{\mathrm{C}}$ from $P_{\mathrm{V}}$, we perturb it with random noise $\epsilon(t)$ to get a noisy image $x_t$ and apply a multi-step denoising process $f_\phi(\cdot)$ using the 2D diffusion prior to obtaining a refined image:

$$\mathbf{I}_\phi(P_{\mathrm{V}}) = f_\phi(x_t; t, y, \hat{\mathbf{I}}_{\mathrm{N}}(P_{\mathrm{V}})), \tag{6}$$

where $y$ is the input text prompt, and $f_\phi$ is a normal-conditioned ControlNet with the Stable Diffusion backbone. We denoise the image from timestep $t$ to a completely clean image $\mathbf{I}_\phi$, which is different from the typical single-step SDS loss in Dreamfusion [2]. The starting timestep $t \in (0, 1)$ gradually decreases from $0.5$ to $0.3$ as training iteration increases, balancing the noise reduction to enhance details without disrupting the original content. This refined image is then used to optimize the texture through the L1 and LPIPS [83] loss:

$$\mathcal{L}_{\text{VG}} = \left\| \hat{\mathbf{I}}_{\text{C}}(P_{\text{V}}) - \mathbf{I}_\phi(P_{\text{V}}) \right\|_1 + \mathcal{L}_{\text{LPIPS}} \left( \hat{\mathbf{I}}_{\text{C}}(P_{\text{V}}), \mathbf{I}_\phi(P_{\text{V}}) \right). \tag{7}$$

While refining the visual textures, we jointly optimize the tactile texture with a **T**actile **G**uidance loss $\mathcal{L}_{\text{TG}}$. Similar to $\mathcal{L}_{\text{VG}}$, we add random noise to a rendered tactile normal map $\mathbf{I}_{\text{T}}(P_{\text{T}})$, generated from a tactile camera pose $P_{\text{T}}$, and then apply a multi-step denoising process to obtain a refined normal image $\mathbf{I}_\psi$. To capture the distribution of tactile normals, we replace the standard diffusion prior with a *Texture DreamBooth*, a customized model created by fine-tuning the Stable Diffusion model $f_\psi(\cdot)$ for each tactile texture using a Low-Rank Adapter (LoRA) [84] with DreamBooth [85]. We leave the LoRAs training details in Appendix A.3. We use the output $\mathbf{I}_\psi$ to refine the tactile texture:

$$\mathcal{L}_{\text{TG}} = 1 - \cos \left( \hat{\mathbf{I}}_{\text{T}}(P_{\text{T}}), \mathbf{I}_\psi(P_{\text{T}}) \right). \tag{8}$$

**Optimization.** The overall loss function is:

$$\mathcal{L} = \lambda_{\text{VM}}\mathcal{L}_{\text{VM}} + \lambda_{\text{TM}}\mathcal{L}_{\text{TM}} + \lambda_{\text{VG}}\mathcal{L}_{\text{VG}} + \lambda_{\text{TG}}\mathcal{L}_{\text{TG}}. \tag{9}$$

To initiate our texture grid, we start by only optimizing the visual matching loss $\mathcal{L}_{\text{VM}}$ and tactile matching loss $\mathcal{L}_{\text{TM}}$ for 150 iterations with $\lambda_{\text{VM}} = 500$ and $\lambda_{\text{TM}} = 1$. After that, we run optimization for another 50 iterations to refine the output guided by diffusion priors. We reduce $\lambda_{\text{TM}}$ from 1 to 0.05, change $\mathcal{L}_{\text{VM}}$ from per-pixel error to mean-color error to allow more flexibility in texture refinement, and add the visual guidance loss $\mathcal{L}_{\text{VG}}$ and tactile guidance loss $\mathcal{L}_{\text{TG}}$ with $\lambda_{\text{VG}} = 5$ and $\lambda_{\text{TG}} = 0.05$.

### 4.3 Multi-Part Textures

Another advantage of our 3D texture field is its ability to easily define non-uniform textures in 3D, allowing different tactile textures to be assigned to distinct parts of an object. For instance, when generating a 3D asset of "a cactus in a pot", we can apply different textures to the cactus and the pot, by incorporating two tactile inputs along with a text prompt that specifies the texture assignment, e.g., "cactus with texture A, pot with texture B".

**Part segmentation based on diffusion features.** Our method can automatically segment object parts based on the text prompt without manual annotation. We leverage the internal attention maps of diffusion models to segment the rendered views $\mathbf{I}_{\text{A}}(P_{\text{V}})$ of the object. Specifically, we add a random noise of level $t$ to $\mathbf{I}_{\text{A}}$ and apply one denoising step with the SD UNet. Inspired by DiffSeg [86] and other text-to-image methods [87, 88], we perform unsupervised part segmentation by aggregating and spatially clustering self-attention maps from multiple layers. To assign part labels, we aggregate cross-attention maps corresponding to different parts and match them with the unlabeled segmentation maps based on Kullback–Leibler (KL) divergence. This results in a list of labeled segmentation masks $\mathbf{M}^n(\mathbf{I}_{\text{A}}) \in \{0, 1\}^{H \times W}$, $n \in \{1, \ldots, N\}$, where $n$ denotes one of the $N$ object parts. For example, given the prompt "cactus in a pot," we aggregate cross-attention maps for "cactus" and "pot" from different layers, then assign each part segmented from the clustered self-attention maps to either cactus or pot according to respective cross-attention maps. More details are included in Appendix A.1.

**3D part label field.** Since the 2D part segmentations may contain noise and lack multi-view consistency, we integrate them from different viewpoints into a coherent 3D part label field. We achieve this by incorporating a part label $s \in \mathbb{R}^N$ into our texture field $\beta(\cdot)$. To supervise it, we render part label maps $\hat{S}(P_{\text{V}})$ from sampled camera poses $P_{\text{V}}$ and compute the cross entropy loss:

$$\mathcal{L}_s = \mathcal{L}_{\text{CE}} \left( \hat{S}(P_{\text{V}}), S(\mathbf{I}_{\text{A}}(P_{\text{V}})) \right). \tag{10}$$

where $\hat{S}(P_{\text{V}})$ is the part logits and $S(\mathbf{I}_{\text{A}}(P_{\text{V}})) \in \{0, 1\}^{H \times W \times N}$ are labels concatenated from $\mathbf{M}^n$.

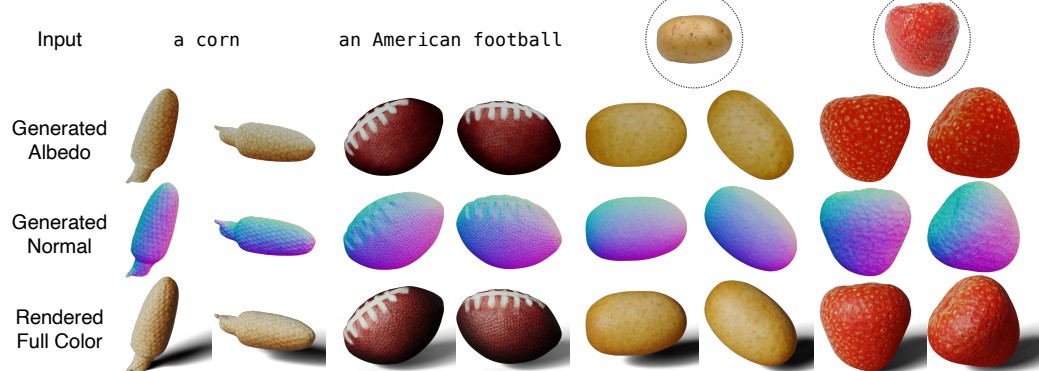

Figure 5: **3D generation with a single texture.** For each object, we show generated albedo (top), normal (middle), and full color (bottom) renderings from two viewpoints. Our method works for both text-to-3D (corn and football) and image-to-3D (potato and strawberry), generating realistic and coherent visual textures and geometric details. (We use roughness=0.5 when rendering color views in Blender for Figures 1, 5, 6, and 7.)

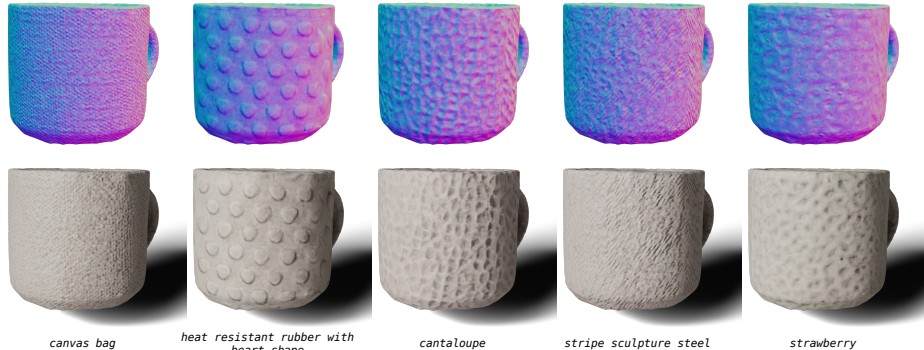

A coffee cup with … texture

Figure 6: **Diverse textures with the same object.** With additional texture cues from tactile data, we can synthesize diverse textures with the same coarse shape for customized designs.

**Optimization with multi-part textures.** To learn multi-part textures, we keep the same $\mathcal{L}_{\text{VM}}$ and $\mathcal{L}_{\text{VG}}$ loss functions while modifying $\mathcal{L}_{\text{TM}}$ and $\mathcal{L}_{\text{TG}}$ to incorporate part-specific tactile supervision:

$$\mathcal{L}_{\text{TM}} = \sum_{n=1}^{N} \left[ 1 - \cos \left( \hat{\mathbf{M}}^n \odot \hat{\mathbf{I}}_{\text{T}}, \hat{\mathbf{M}}^n \odot \mathbf{I}_{\text{T}}^n \right) \right], \tag{11}$$

$$\mathcal{L}_{\text{TG}} = \sum_{n=1}^{N} \left[ 1 - \cos \left( \hat{\mathbf{M}}^n \odot \hat{\mathbf{I}}_{\text{T}}, \hat{\mathbf{M}}^n \odot \mathbf{I}_{\psi}^n \right) \right], \tag{12}$$

where $\odot$ denotes the Hadamard product, $\mathbf{I}_{\text{T}}^n$ is the rendered reference view using the $n$-th part's tactile data, $\mathbf{I}_{\psi}^n$ is the refined tactile normal generated by Texture Dreambooth trained on the $n$-th part texture, and $\hat{\mathbf{M}}^n$ is the binary mask for the $n$-th part obtained from $\hat{S}$ rendered from our learned 3D label field. We omit $P_{\text{T}}$ for clarity since all patches are rendered from the same tactile camera pose.

## 5 Experiment

We present comprehensive experiments to verify the efficacy of our method. We perform qualitative and quantitative comparisons with existing baselines and ablation studies on the major components.

**Dataset.** We have collected 18 diverse types of textures from daily objects in our `TouchTexture` dataset, five tactile patches per texture, and pair them with a short description. Please see Appendix A.2 for the full list. We randomly sample one patch to initialize the tactile UV map with the image quilting algorithm [77] and use five patches to train the customized Texture DreamBooth. For base mesh generation given a text or an image input, while any mesh generation method is applicable,

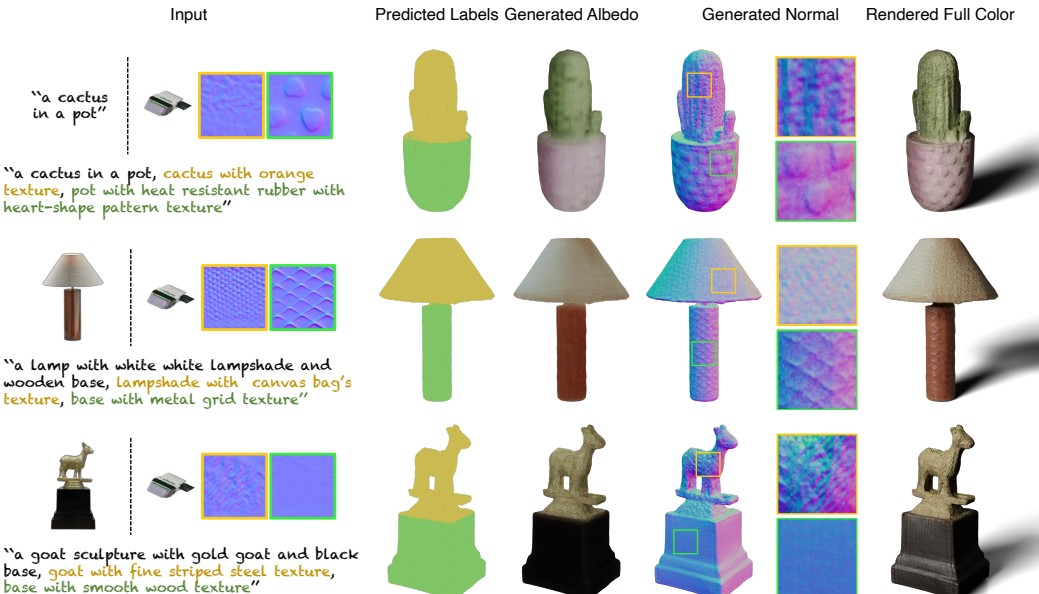

Figure 7: **Multi-part texture generation.** Our method allows users to specify an object (via text or image) and its two parts to assign different textures (color-coded text prompts correspond to text descriptions for two textures). We show paired results for the predicted label, albedo, normal, and full-color renderings. The zoom-in patches demonstrate the generated normal textures on different parts.

we use Wonder3D [13] in the main experiments but include results using InstantMesh [89] and RichDreamer [90] in Appendix A.4 as well. Specifically, Wonder3D takes an image as input and outputs a mesh with albedo stored as vertex colors. We then convert the vertex albedo into an albedo UV map as described in Section 4.1. During optimization in Section 4.2, we refine the albedo and tactile textures with the "textured prompts", i.e., *A [object name] with V\* texture*, where *V\** represents the text description corresponding to the selected tactile texture map.

**3D generation with a single texture.** Figure 5 shows our 3D generation results for a single texture, showing albedo, normal, and full-color rendering from two viewpoints for each object. Our method works for both text input (the corn and the football) and image input (the potato and the strawberry), generating realistic, coherent, and high-resolution visual and geometry details. Figure 6 shows the results of applying different textures to the same object (a coffee cup), with normal rendering on top and color rendering below. Our joint optimization produces coherent visual-tactile textures and smooth, natural shadows from geometric variations.

**Multi-part texture generation.** As introduced in Section 4.3, users can specify an object (via text or image) and assign different textures to two distinct parts. Figure 7 shows results for multi-part texture synthesis. The left column shows the text or image input as well as the tactile input. The image input can be either real or generated from a text prompt. The color-coded text prompts correspond to text descriptions for two textures. The right columns show the results of the albedo, normal, and full-color rendering. Our method effectively segments parts with our 3D label field. The joint optimization successfully applies the textures to each corresponding part designated by the user and generates an overall coherent visual appearance and tactile geometry.

**Baselines.** To our knowledge, this work is the first to leverage high-resolution tactile sensing to enhance geometric details for 3D generation tasks. Thus, we compare our method with existing text-to-3D and image-to-3D methods. For text-to-3D, we compare with DreamCraft3D [22] and use the same textured prompt, i.e., a prompt with a text description of the target tactile texture, as input. DreamCraft3D focuses on geometry sculpting using neural field and DMTet [91], followed by texture boosting through fine-tuning DreamBooth with augmented multi-view rendering, requiring 10x computational cost compared to our method. For image-to-3D, we compare with DreamGaussian [49] and Wonder3D [13]. DreamGaussian employs fast generation using 3D Gaussian primitives and refines an albedo map in Stage 2 with a multi-step denoising MSE loss. Wonder3D generates paired multi-view RGB and normal images, using a geometry fusion process to generate a 3D result. The input images remain the same as ours for these two baselines. We use the official implementations for all baselines.

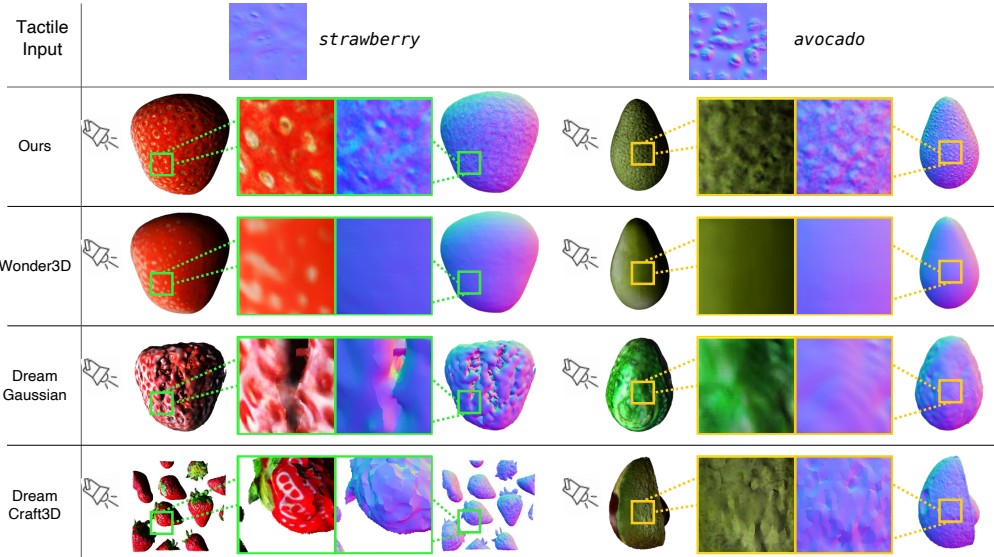

Figure 8: **Baseline comparison.** Compared to the SOTA image-to-3D (Wonder3D and DreamGaussian) and text-to-3D (DreamCraft3D) baselines, our method produces significantly more plausible low-level geometry. For a fair comparison, we use the same input image for the first three rows.

Table 1: **Human perceptual study.** For all paired comparisons, our method is preferred ($\geq 50\%$) over the baselines for both texture appearance and geometric details.

|  | Ours vs DreamGaussian | Ours vs Wonder3D | Ours vs DreamCraft3D |
|---|---|---|---|
| **Texture** | $85.43 \pm 4.76$ | $86.36 \pm 3.93$ | $61.54 \pm 4.43$ |
| **Geometry** | $92.85 \pm 3.47$ | $88.07 \pm 3.80$ | $84.20 \pm 4.17$ |

**Qualitative evaluation.** Figure 8 shows qualitative results of our method compared against three baselines. For each example, we show color and normal rendering with zoomed-in patches at the same location for detailed comparison. Our method achieves higher visual fidelity, more realistic details in normal space, and better color-geometry alignment than the baselines. In particular, our textures exhibit sharper details than those of Wonder3D, which tend to be overly smooth. In contrast to DreamGaussian and DreamCraft3D, our generated geometry details are more realistic and align well with the color appearance. In the avocado example, DreamCraft3D suffers from the "Janus problem", generating a brown core on both sides.

**Quantitative evaluation.** We perform a human perceptual study using Amazon Mechanical Turk (AMTurk). We set up a paired test, showing a reference prompt and two rendering results, one generated with our method and the other generated with one of the baselines. We conduct two separate surveys to evaluate the texture appearance and geometric details, respectively. For texture appearance, we render full-color RGB images and ask users "Which of the following views has more realistic textures?". For geometric details, we render shaded colorless images that only demonstrate the mesh geometry and ask users "Which of the following views has more realistic geometric details?". Please see Appendix A.5 for example screenshots of the paired rendering. Each user has two practice rounds followed by 20 test rounds to evaluate our method against DreamGaussian, Wonder3D, and DreamCraft3D. All samples are randomly selected and permuted, and we collect 1,000 responses.

Table 1 shows the mean and standard deviation of users' preference score; our method is preferred over all baselines. While DreamCraft3D has a close performance regarding texture appearance, it fails to obtain high-resolution geometric details that align well with color texture.

**Ablation study.** We perform an ablation study to evaluate key aspects of our method. We render both front and back views of a sampled object for each experiment. To illustrate details and cross-modal alignment, we present a global full-color view on the left, a global normal view on the right, and patch views of the full-color, albedo, and normal renderings in between. Figure 9 shows results of ablating diffusion-based guidance losses for texture refinement, specifically the visual guidance $\mathcal{L}_{\text{VG}}$ and tactile guidance $\mathcal{L}_{\text{TG}}$.

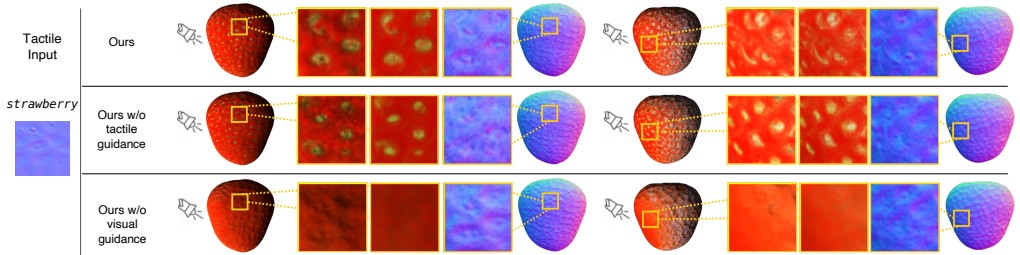

Figure 9: **Ablation study regarding texture refinement**. We ablate our method regarding the tactile guidance loss $\mathcal{L}_{TG}$ and visual guidance loss $\mathcal{L}_{VG}$. Removing $\mathcal{L}_{TG}$ results in fewer details of the generated tactile texture. Removing the visual guidance introduces misaligned visual and tactile normal textures. For example, the bumps in the normal map are misaligned with the locations of white seeds in the albedo rendering, as shown in the zoomed-in patches. Our method encourages the generated color to be consistent with the tactile normal using ControlNet-guided visual refinement loss while also enhancing the details in the tactile texture.

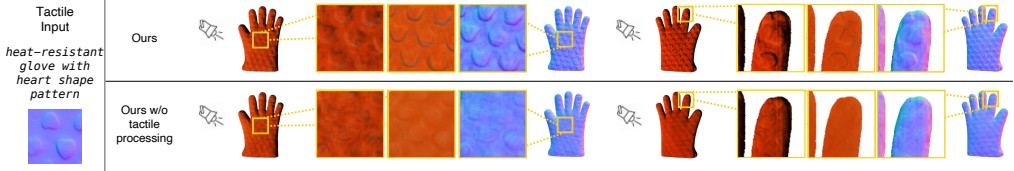

Figure 10: **Ablation study regarding tactile preprocessing**. Without tactile data preprocessing including high-pass filtering and contact area cropping, the generated geometric details tend to be flat and unrealistic.

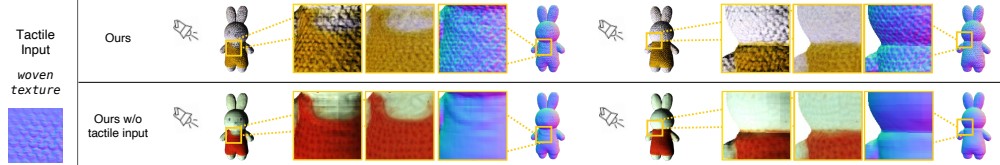

Figure 11: **Ablation study regarding tactile input**. We remove the tactile input while keeping the refinement loss. Without the tactile information, the generated 3D assets fail to capture fine-grained geometric details.

Omitting $\mathcal{L}_{TG}$ reduces tactile texture details, while omitting $\mathcal{L}_{VG}$ introduces misalignment between visual and tactile normal textures; for instance, bumps in the normal map do not match the white seeds in the albedo rendering. Our method ensures color consistency with tactile normals via ControlNet-guided visual refinement and enhances tactile texture details.

We also study the efficacy of tactile data processing by comparing our method with the texture map synthesized using the original tactile data without preprocessing stated in Section 3. Figure 10 shows that using the original data produces much more flattened textures since the low-frequency deformation of the gel pad due to uneven contact during the data collection would dominate the tactile signal and thus degrade the details of synthesized texture maps. Figure 11 shows the results of removing tactile input and only optimizing the albedo map with "textured prompts" using $\mathcal{L}_{VM}$ and $\mathcal{L}_{VG}$. Without tactile input, the output mesh is overly smooth, demonstrating the insufficiency of using text prompts only for fine geometry texture generation.

## 6 Discussion

Recent methods in 3D generation often struggle with unrealistic geometric details. To address this, we have introduced a novel approach that incorporates tactile information. Our method synthesizes both visual and tactile textures using a 3D texture field. Additionally, we have introduced a multi-part texturing pipeline for controllable region-wise texture generation. To our knowledge, this is the first use of high-resolution tactile sensing to improve 3D generation. Our method produces realistic, fine-grained geometric textures while maintaining accurate visual-tactile alignment.

**Limitations.**  The quality of the generated coarse shape depends on the existing 3D generative models, which struggle with complex geometry. Additionally, slight seams may appear in our results due to UV unwrapping.

## Acknowledgments and Disclosure of Funding

We thank Sheng-Yu Wang, Nupur Kumari, Gaurav Parmar, Hung-Jui Huang, and Maxwell Jones for their helpful comments and discussion. We are also grateful to Arpit Agrawal and Sean Liu for proofreading the draft. The project is partially supported by the Amazon Faculty Research Award, Cisco Research, and the Packard Fellowship. Kangle Deng is supported by the Microsoft Research PhD Fellowship. Ruihan Gao is supported by the A*STAR National Science Scholarship (Ph.D.).

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

# Appendix

## A   Implementation Details and Additional Results

Please check out our webpage for video results and more examples.

### A.1   Diffusion-based Multi-Part Segmentation

In Section 4.3, we have mentioned using attention maps of the diffusion process to segment the input image based on text prompts. We provide more details here. Assume an input image $x$ and its corresponding prompt $y = [y_1, y_2, \cdots, y_T]$ are given, where $T$ is the number of tokens from the text encoder ($T = 77$ for CLIP text encoder [92]). Among these, $y_{p_1}, \cdots, y_{p_N}$ define $N$ parts in the image $x$, where $p_1, \cdots, p_N$ indicate the indexes of the token. For example, with the prompt "a cactus in a pot", the second token $y_2$ ($p_1 = 2$) "cactus" and the fifth token $y_5$ ($p_2 = 5$) "pot" correspond to two parts ($N = 2$) of the object in the image, and our method outputs the segmentation mask for each part. Specifically, we first perturb $x$ using random noise $\epsilon_t$ with a noise level of $t$, which is empirically set as 0.2. We then use the Stable Diffusion UNet to denoise $x_t$ for one step. During the denoising process, we collect the cross-attention and self-attention probability maps for each of the 16 layers.

**Segment and create masks using self-attention layers.**   Following DiffSeg [86], we aggregate the 16 self-attention probability maps to a single map $\mathcal{A}_f$ with the shape of $64 \times 64 \times 64 \times 64$, and run iterative attention merging [86] on upsampled $\mathcal{A}_f$ to obtain a preliminary cluster probability map $\tilde{\mathcal{A}}_f \in \mathbb{R}^{512 \times 512 \times K}$, where each $\tilde{\mathcal{A}}_f[:,:,k], k = 1, \cdots, K$, is a probability map,

$$\sum_{i,j} \tilde{\mathcal{A}}_f[i,j,k] = 1, \quad k = 1, 2, \cdots, K. \tag{13}$$

We can then obtain a preliminary segmentation mask $\tilde{S}$,

$$\forall i, j, \qquad \tilde{S}[i,j] = \arg\max_k \tilde{\mathcal{A}}_f[i,j,k]. \tag{14}$$

**Aggregate labels with cross-attention layers.**   Similarly, we aggregate the 16 cross-attention maps and up-sample them to $\mathcal{A}_c$ with the shape of $512 \times 512 \times 77$. To find the exact pixels corresponding to each part specified in the prompt input, we extract cross-attention maps associated with the $N$ tokens $y_{p_1}, \cdots, y_{p_N}, \tilde{\mathcal{A}}_c \in \mathbb{R}^{512 \times 512 \times N}$,

$$\forall i, j, n \qquad \tilde{\mathcal{A}}_c[i,j,n] = \frac{\mathcal{A}_c[i,j,p_n]}{\sum_{i',j'} \mathcal{A}_c[i',j',p_n]}. \tag{15}$$

Silimar to the self-attention probability maps $\tilde{\mathcal{A}}_f$, here we normalize the cross-attention maps over the spatial dimensions to ensure $\tilde{\mathcal{A}}_c[:,:,n]$ is a probability map for each $n$, where $\sum_{i,j} \tilde{\mathcal{A}}_c[i,j,n] = 1$.

**Associate masks with labels.**   Note that $\tilde{S}$ contains $K$ parts but in general $K \neq N$. Therefore, we need to assign each of the $K$ segments to one of the $N$ parts specified in the prompt.

We calculate a KL divergence matrix $D \in \mathbb{R}^{K \times N}$ between $\tilde{\mathcal{A}}_f$ and $\tilde{\mathcal{A}}_c$,

$$D[k,n] = D_{\mathrm{KL}}\left(\tilde{\mathcal{A}}_f[k] \,\|\, \tilde{\mathcal{A}}_c[n]\right), \quad k = 1, 2, \cdots, K, \quad n = 1, 2, \cdots, N. \tag{16}$$

We merge the preliminary mask $\tilde{S}$ according to $D$ to get the final segmentation $S$,

$$\forall i, j, \qquad S[i,j] = \arg\min_n D[\tilde{S}[i,j], n]. \tag{17}$$

An example of output masks is shown in Figure 12.

### A.2   Dataset Details

Figure 13 shows additional textures included in our `TouchTexture` dataset, and Table 2 shows the list of text descriptions corresponding to each texture.

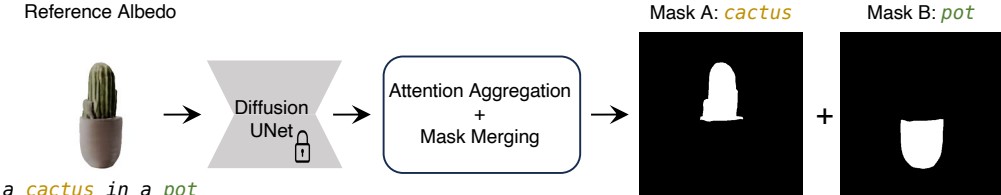

Figure 12: **Illustration of diffusion-based multi-parts segmentation.** At each iteration, we run a forward pass of the diffusion model for the target albedo image, aggregate its self-attention and cross-attention maps, and compute KL distance to merge the masks into $N$ parts. We show an example of "a cactus in a pot", where we extract the masks corresponding to two tokens "cactus" and "pot", shown as *Mask A* and *Mask B*. The segmentation masks are then used to supervise the label field training to enable multi-part synthesis.

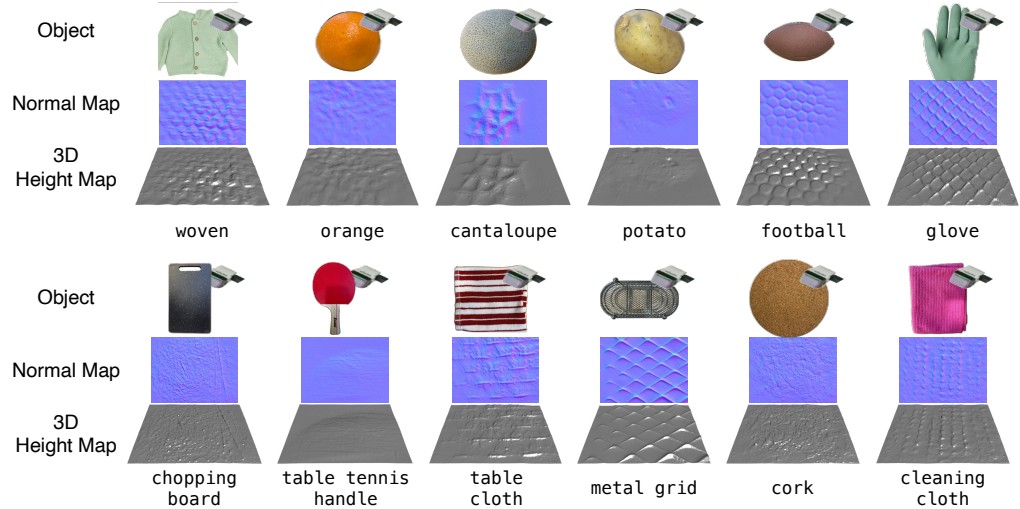

Figure 13: **Additional materials in tactile normal dataset** `TouchTexture`. We collect tactile normal data from 18 daily objects featuring diverse tactile textures. We show the tactile normal map and a 3D height map for each object. This library of tactile samples covers a wide range of diverse materials commonly found in daily life.

## A.3   Training Details

Regarding the network, we use TCNN encoding with two base linear layers, which then branches out into three output layers, one linear layer followed by sigmoid activation for albedo output, one linear layer followed by $\tanh$ activation for tactile normal output, and one optional linear layer for label field. We follow DreamBooth [85] to train texture LoRAs with Stable Diffusion (SD) V1.4 for the tactile guidance loss. We find empirically that SD V1.4 works better for generating tactile normal maps. We use ControlNet (v1.1 - normalbae version) with SD V1.5 for the diffusion loss. We follow Wonder3D to generate the base mesh and train the texture field network using Adam optimizer with $lr = 0.01$. We train all models on A6000 GPUs and each experiment takes about 10 mins and 20G RAM to run.

## A.4   Flexible Base Mesh Generation

Given a text or image input, our method seamlessly integrates with a wide range of mesh generation approaches to enhance the 3D output with refined details. Here, we presents results using RichDreamer [90] for base mesh generation in Figure 14a and results using InstantMesh [89] in Figure 14b. RichDreamer generates detailed and consistent 3D meshes from text prompts by incorporating depth and normal priors during pre-training, while InstantMesh employs a feed-forward image-to-3D approach with a transformer-based reconstruction model and FlexiCubes for efficient mesh production. Both methods provide solid base meshes, and our approach effectively adapts to each, refining the output meshes with intricate texture details for greater realism and fidelity.

Table 2: List of objects in `TouchTexture` and their text descriptions.

| Object Name | Description |
|---|---|
| avocado | "avocado skin's " |
| strawberry | "strawberry" |
| canvas bag | "canvas bag" |
| striped steel | "fine striped steel" |
| corn | "corn" |
| rubber | "heat-resistant rubber with heart-shape pattern" |
| woven | "woven crochet" |
| orange | "orange" |
| cantaloupe | "cantaloupe skin's" |
| potato | "potato" |
| football | "football" |
| glove | "thin rubber with grid pattern" |
| chopping board | 'cutting board" |
| table tennis handle | "smooth wood" |
| table cloth | "patterned table cloth" |
| metal grid | "metal grid" |
| cork | "cork mat" |
| cleaning cloth | "cleaning cloth's" |

### A.5   Human Perceptual Study Details

We perform a human perceptual study on Amazon Mechanical Turk (AMTurk). We set up a paired test, showing a reference prompt and two rendering results, one generated with our method and the other generated with one of the baselines. For each result, we include a single view of the entire object with a zoom-in patch in the red box to show the details. We use directional light shooting in from the left side for all mesh renderings. We give a text prompt, e.g., "an avocado", and ask the user to select the result that better matches the prompt. We render shaded colorless images for geometry evaluation as shown in Figure 15a and render full-color images for texture evaluation as shown in Figure 15b. The estimated time for each test is about 5 minutes, and we pay users a corresponding amount of compensation higher than the minimum wage in the country of the data collector. The online survey does not pose any explicit potential risks expected to be incurred by participants.

## B   Soceital Impacts

Our method of incorporating tactile information into 3D generation enhances the geometric details of the generated results. This advancement helps automatically create high-fidelity assets, which are highly applicable in game and film production. However, there is also a potential risk of misuse, as 3D assets could be used to generate fake content for misinformation. Despite this concern, we believe humans can currently distinguish our synthesized objects from real ones.

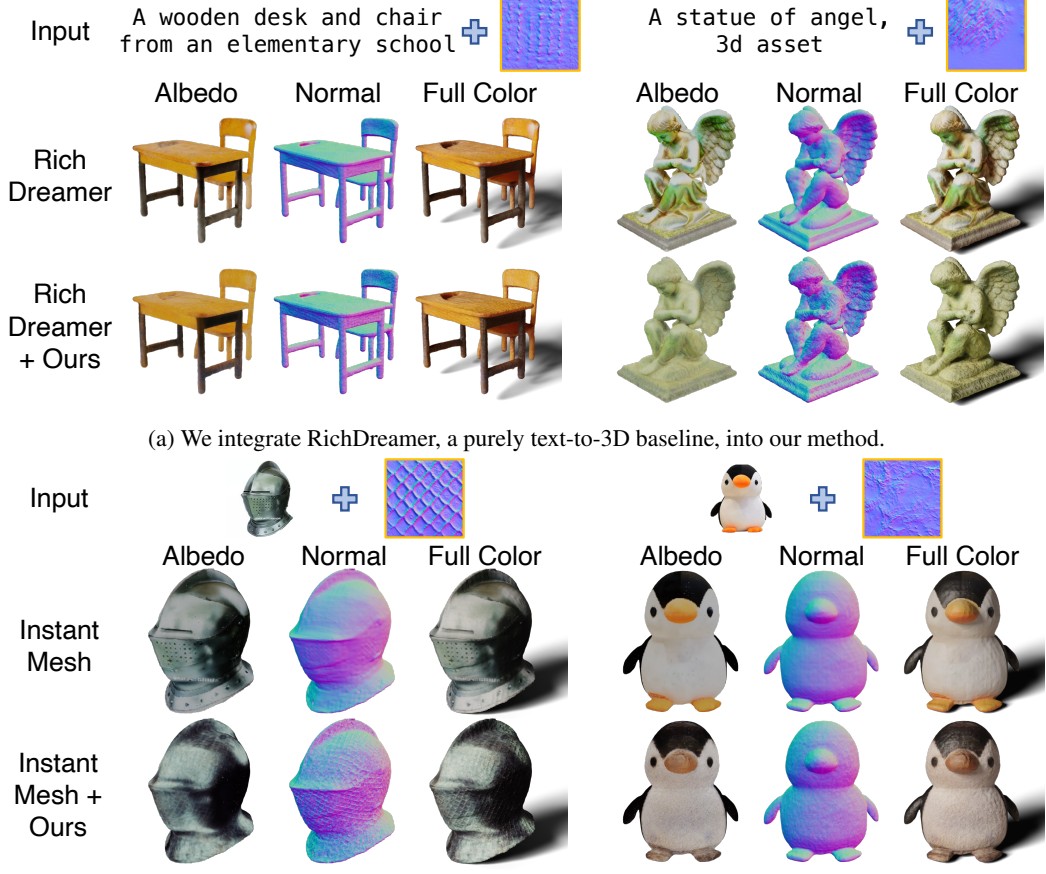

(a) We integrate RichDreamer, a purely text-to-3D baseline, into our method.

(b) We integrate InstantMesh, an image-to-3D baseline, into our method

Figure 14: Our method is flexible and compatible with diverse text-to-3D and image-to-3D pipelines.

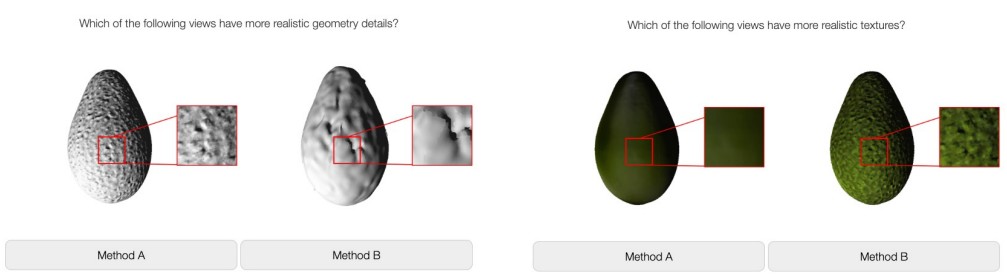

(a) A sample test page for geometry evaluation.  (b) A sample test page for texture evaluation.

Figure 15: **AMTurk setup for evaluation**. For each paired result, we include a single view of the entire object with a zoom-in patch in the red box to show the details. For each result, we include a single view of the entire object with a zoom-in patch in the red box to show the details. We use directional light shooting in from the left side for all mesh renderings. We give a text prompt, e.g., "an avocado", and ask the user to select the result that better matches the prompt.

