# OpenReview forum: "Tactile DreamFusion: Exploiting Tactile Sensing for 3D Generation"
_NeurIPS.cc/2024/Conference — NeurIPS 2024 poster_

### Official Review · Reviewer_a9oZ · 2024-06-26

**Soundness:** 2
**Presentation:** 3
**Contribution:** 3
**Rating:** 7
**Confidence:** 4

**Summary:**

This paper proposes to use the tactile sensing signal (height map and normal map) to improve the 3D generation quality, especially the geometric details. The authors use a 3D mesh generation guided by a normal-conditioned ControlNet to ensure the consistency between the  visual textures and the tactile textures. They also develop a multi-part editing pipeline to generate objects with different texture parts. Experiments and ablation studies demonstrate the effectiveness of the proposed method.

**Strengths:**

The origniality of the paper is good. To my knowledge, this is the first work to use the tactile sensing for 3D generation. The generation quality is satisfying. The generated meshes include good geometric details, which align well with the input tactile signals. The paper is well written and structured. The visualizations clearly demonstrate the quality of the generated meshes.

**Weaknesses:**

The connection between the tactile sensing and the 3D generation is not strong and critical. The tactile signals are only used to generate the normal maps of certain textures. It may be replaced by simpler alternatives such as texture reterival from a normal map database. It needs further elaboration why this combination is necessary.

**Questions:**

1. What is the generation time of the multi-part geneation?
2. Can the authors add an experiment of real image to 3D mesh with real corresponding tactile signals?

**Limitations:**

Limitaions have been addressed in the paper, which include complex geometry generation and slight seams.

---

> ### Author Rebuttal · Authors · 2024-08-07
>
> Thanks for your time and valuable comments. We address the questions below:
>
> ### **Using texture reterival from a normal map database**
>
> We agree with the reviewer that our work can be potentially extended to other high-resolution geometrical texture data such as the one from a normal map database. In this work, we focus on the data acquired from tactile sensors, due to several reasons. First, the normal map database could be limited in terms of texture variation, e.g., different avocados, as it often contains a single or very few normal maps per object. Second, the normal map data is sometimes hand-crafted rather than captured from real data, making it less realistic. Third, when the user has a particular texture in mind to synthesize, e.g., customizing the plush toy with the fabric of their favorite sweater. In these regards, we believe high-resolution tactile sensors provide a quick and scalable way to capture the precise textures of real objects and, therefore, will be a common way to capture surface textures, in the near future.
>
> Moreover, in the future, we can extend the work of using tactile feedback to model the physical properties, such as the hardness of the material, as part of the object rendering. We thank the reviewer for pointing this out and will add this discussion in the revision.
>
> ### **Generation time of the multi-part generation**
> As mentioned in the supplementary material, we train all models on A6000 GPUs and each experiment takes about 10 mins and 20G VRAM to run. To be specific, single-part experiment takes about 6 mins and multi-part experiment takes about 8 mins to run.
>
> ### **Experiment of using real Image and real corresponding tactile as input**
> We show one example result of using a real image and corresponding captured tactile signals as input in Figure 3 of the attached PDF. Our method can recover the color and geometric details reasonably well.

---

> > ### Comment · Reviewer_a9oZ · 2024-08-12
> >
> > Thank you for the response. The explaintation is reasonable to me.

---

### Official Review · Reviewer_ecYi · 2024-07-10

**Soundness:** 2
**Presentation:** 2
**Contribution:** 1
**Rating:** 4
**Confidence:** 4

**Summary:**

This paper proposes a lightweight 3D texture field that ensures the consistency between visual and tactile textures while preserving photorealism. The experiments demonstrate that quantitative and qualitative results show good generation quality.

**Strengths:**

1. The authors pioneered the use of tactile sensing to enhance geometric details.
2. The exposition is good. The paper is easy to understand.
3. They created a TouchTexture dataset, comprising 16 everyday objects, contributing a new dataset to the community.

**Weaknesses:**

1. This paper suggests that existing methods struggle to achieve fine-grained geometric details. However, methods like Neuralangelo and PermutoSDF can recover highly detailed geometric information.

2. The TouchTexture dataset presented in Figure 3 and the supporting materials seemingly do not capture the "local geometric intricacies."

3. There are only very limited objects been shown in the main paper and supplementary material. Are those objects cherry-picked? It would be great if more results on in-the-wild object can be provided to show the generalisation ability of the model.

4. Ablation Study is an important part in paper and it would be more convincing with both quantitative and qualitative experiments, and cannot be simply summarized with only several sentences. If there are any figures and tables in the article, please indicate the specific table number or figure number in the analysis and analyze according to the specific visualization results.

- [Neuralangelo: High-Fidelity Neural Surface Reconstruction. CVPR2023]
- [PermutoSDF: Fast Multi-View Reconstruction with Implicit Surfaces using Permutohedral Lattices. CVPR2023]

**Questions:**

See my previous sections.

**Limitations:**

I'm not confident about voting for accepting this paper because of the potential similarity to existing methods and the limited novelty.
There is more engineering effort than novelty. The novelty might be limited.

---

> ### Author Rebuttal · Authors · 2024-08-07
>
> Thanks for your questions and comments. We would like to clarify that in this work, our main contribution is leveraging tactile sensing to enhance geometric details for 3D generation tasks. (1) We are the first to leverage tactile sensing to synthesize high-fidelity geometric details for 3D generation. (2) We present a new technique to learn high-quality 3D texture fields with aligned visual and geometry textures, with a multi-part extension that allows us to synthesize multiple textures across various regions. (3) Our experiments show that our method outperforms existing text-to-3D and image-to-3D methods.
>
> ### **Comparison with the different setup of reconstruction pipeline**
> We agree that methods like Neuralangelo (Li, et al., 2023) and PermutoSDF (Rosu, et al., 2023) can recover highly detailed geometric information. However, these methods focus on 3D reconstruction tasks and require sufficient data to achieve high-resolution reconstruction, e.g.,  “50 images” for an object and “~200-1000 images” for a larger scene. In our setup, we focus on a different task of text-driven 3D generation. We make use of a similar 3D representation, i.e., a multi-resolution 3D hash grid, as a texture field and strive to generate highly detailed 3D assets with minimal input, a text prompt / single-view image and a single tactile patch.
>
> ### **Local geometric intricacies captured by Tactile data**
> The GelSight mini sensor we use to collect the TouchTexture dataset has a sensing area of 21mm x 25mm. The normal maps and height maps in Figure 3 of the main paper and the supporting materials show the tactile data for a single touch, cropped to 18mm x 24mm in physical scale and 240 x 320 in pixels, which corresponds to a tiny patch as shown in Figure 2 of the main paper. We refer to the “submillimeter scale” geometry captured by the sensor as “local geometric intricacies”.
>
> ### **More Diverse Results**
> We show more complex and diverse results in Figure 4 and Figure 5 of the attached PDF. Our method is compatible with most pipelines that generate colored meshes, and can enhance the geometric details of their output. With different backbones such as RichDreamer (Qiu et al., 2024) and InstantMesh (Xu et al., 2024), our method is able to generate diverse, in-the-wild objects. These results showcase the generalizability of our method. We will include more examples in the revised draft.
>
> ### **Ablation studies**
> In the main paper, Figures 9,10, and 11 illustrate the ablation studies and the analysis is provided in Line 238-250. Directly applying a normal texture map to a base mesh without joint optimization may introduce unnatural appearance, as the additional normal texture could be conflicting with the original albedo map, as shown in the misaligned strawberry seeds in Figure 9. Figure 10 shows that using the raw tactile data without proposed preprocessing produces much more flattened textures. This is because low-frequency deformation of the gel pad would dominate the tactile signal, reduce the signal-to-noise ratio, and degrade the synthesized geometric details. We also ablate the tactile input, and the example results are shown in Figure 11. Removing tactile input produces overly smooth meshes, as a text prompt could not provide sufficient guidance to geometric details at the millimeter scale. Figure 2 in the attached PDF also shows the ablation of tactile loss on a part of the object, demonstrating that the diffusion prior can infer some geometric variation to a certain level but is not capable of generating high-fidelity regular and consistent textures. We are happy to conduct additional ablation studies if there are specific aspects the reviewers would like us to investigate.

---

> > ### Comment · Reviewer_ecYi · 2024-08-12
> >
> > Thank you for taking time and preparing a rebuttal which I read carefully.

---

### Official Review · Reviewer_MUQX · 2024-07-12

**Soundness:** 2
**Presentation:** 3
**Contribution:** 3
**Rating:** 6
**Confidence:** 4

**Summary:**

This submission addresses the long-standing challenge of enhancing geometric details in results produced by text-to-3D and image-to-3D pipelines. The approach introduces a novel method that leverages tactile normal modality to synthesize high-fidelity geometric details. Additionally, it employs attention maps during the diffusion process to segment input images based on text prompts, allowing for the synthesis of multiple textures across various regions. The results demonstrate that this method effectively recovers geometric details and ensures alignment between geometry and color.

**Strengths:**

1. The approach is innovative in utilizing tactile normal modality to enhance geometric details.
2. The introduction of a newly collected tactile dataset, TouchTexture, is beneficial to the research community.

**Weaknesses:**

1. The paper claims compatibility of the proposed method with both text-to-3D and image-to-3D pipelines. However, DreamCraft3D, selected as a text-to-3D pipeline, requires both an image and a text caption as inputs. Although DreamCraft3D can be considered as a 'text-to-image, text & image-to-3D' process, it differs from a purely text-to-3D approach. Thus, the compatibility between the proposed method and purely text-to-3D pipelines remains questionable.

2. The pipeline overview in Figure 4 indicates the need for a reference image and tactile input. The paper does not address how to select an appropriate tactile input for the reference image, nor how to ensure the tactile details are compatible with the object in the image.

3. The paper presents an intriguing text-guided segmentation strategy that leverages attention maps during the diffusion process based on text prompts. However, lacking expertise in diffusion-based segmentation, I am unfamiliar with the efficiency and success rate of this method, but I am positively impressed by it.

4. The generalization and diversity of the proposed method are also of concern. The objects presented in the results are not very complex, and the dataset comprises 16 popular categories. It remains unclear how well the method would perform on more complex objects, such as those in sci-fi or fantasy genres.

**Questions:**

1. I am curious about your criteria for selecting 3D generation baselines. For image-to-3D tasks, to my knowledge, more advanced baselines such as 'InstantMesh: Efficient 3D Mesh Generation from a Single Image with Sparse-view Large Reconstruction Models' and 'TripoSR: Fast 3D Object Reconstruction from a Single Image' offer superior quality and may yield stronger results. For text-to-3D tasks, 'RichDreamer: A Generalizable Normal-Depth Diffusion Model for Detail Richness in Text-to-3D' is a noteworthy pure text-to-mesh method.

2. The proposed method in the paper first refines normals and then optimizes colors based on the refined normals. Does this imply that the geometric details introduced by your method are not derived from the original colors (i.e. not very necessary to original colors) of the objects? Additionally, could the color optimization process potentially disrupt the alignment between the original colors and the refined colors, potentially failing to meet the initial requirements?

3. I am curious about the generalization and diversity capabilities of the proposed method. Can it handle more complex objects? Furthermore, with more powerful baselines, can this method achieve higher quality results on more complex cases?

**Limitations:**

The authors acknowledge their limitations in the main paper and address potential social impacts in the supplementary materials. Regarding the first limitation, the implementation of new 3D generative models, such as 'Unique3D: High-Quality and Efficient 3D Mesh Generation from a Single Image' and 'Era3D: High-Resolution Multiview Diffusion using Efficient Row-wise Attention', is recommended. For the second limitation, utilizing a more powerful computer graphics tool could be beneficial. Concerning the social impact, issues related to deepfakes and the potential for misinformation are noteworthy. The authors assert that humans can currently distinguish their synthesized objects from real ones, a claim with which I concur. Although it is of low priority, it would be preferable for the authors to include a comparison between a generated object and a real one; a single case would suffice.

---

> ### Author Rebuttal · Authors · 2024-08-07
>
> Thank you for your encouraging comments and feeback. We answer each question as below:
>
> ### **Is the proposed method compatible with a purely text-to-3D pipeline?**
> Yes, our method is compatible with purely text-to-3D backbones, such as RichDreamer (Qiu, et al., 2024). In our method, we generate our base mesh by first generating an image from the input text prompt using SDXL (Podell, et al., 2023) and then running Wonder3D (Long, et al., 2023), which is a ‘text-to-image-to-3D’ pipeline. However, our method is also compatible with most pipelines that output colored meshes. As suggested, we used a state-of-the-art text-to-3D method RichDreamer, and integrated it into our method. Specifically, we generate the base mesh using RichDreamer, and finetune the albedo and normal UV map using our proposed tactile matching loss, diffusion-based refinement loss, and regularization loss. Example results of both RichDreamer and our full method are shown in Figure 4 in the attached PDF.  As shown in Figure 4, our method is able to generate 3D objects corresponding to the text prompt while adding high-fidelity geometric details from tactile inputs.
>
> ### **More advanced image-to-3D baseline and more complex objects**
> Similarly, we can integrate InstantMesh (Xu, et al., 2024), a more recent image-to-3D method, into our method. Figure 4 and Figure 5 in the attached PDF demonstrate the results for more complex objects using the new backbones of RichDreamer and InstantMesh. Our method enhances the geometric details since the tactile information provides guidance of finer scale than the resolution of the base mesh generated by the backbones. These results highlight the compatibility of our method with different backbones, and its generalizability to complex objects. We will include them in the revision.
>
> ### **How to select tactile input and how to ensure the tactile details are compatible with the object?**
> The tactile input can be selected either for realism or creativity. For generating realistic outputs, we would choose the tactile input similar to the object, e.g., using a tactile patch collected from real strawberry to generate a strawberry mesh. Otherwise, if we aim for creativity, we can choose any tactile texture we want, as the various coffee cups show in Figure 6 of the main paper.  Since the tactile details are of millimeter scale, they are added upon the coarse geometry as a normal UV map, which is compatible with the base mesh. To further ensure alignment between the color and tactile details, we jointly optimize the albedo and normal maps using the refinement loss.
>
> ### **What are the efficiency and success rate of the attention-based segmentation?**
> Since we leverage the same diffusion model to compute the attention maps for segmentation, it is more memory efficient than using  additional components such as off-the-shelf segmentors like SAM. In terms of running time, a single-part experiment takes about 6 mins and a multi-part experiment takes about 8 mins.
>
> To quantitatively evaluate our segmentor, we manually segment 3 meshes to obtain ground-truth segmentation masks. We then run our segmentor on 100 renderings for each mesh. In terms of the metric, we calculate the IoU between our predicted masks and the labeled masks and also compute the accuracy of predicted labels. Our diffusion-based segmentor reaches an average IoU of 0.588 and an accuracy of 83.7%, which provides sufficient segmentation for multi-part optimization. Note that our multi-part optimization can partially resolve inaccurate and inconsistent segmentation by aggregating the gradients from different views, so our 2D segmentation does not need to be perfect. We are happy to expand this evaluation in the revision.
>
> ### **Clarification about the color and geometry optimization**
> The coarse geometry is fixed after the base mesh generation while the local geometric details, defined by normal UV maps, are further refined using tactile information, and thus not derived from the original colors. The color is also optimized simultaneously with the normals using a diffusion-based refinement loss. The optimization could introduce changes of color to align visual and tactile modality, while we add the regularization term so that the refined color maintains consistent with reference color on a larger scale. We balance the loss weight and a single set of parameters works for all our experiments.
>
> ### **Comparison with real objects**
> We show one example result of using a real image and corresponding captured tactile signals as input in Figure 3 of the attached PDF. Although our method can recover the color and geometric details reasonably well, we think humans can still distinguish between the real and generated objects by comparing with the input real image.

---

> > ### Comment · Reviewer_MUQX · 2024-08-13
> >
> > After reading the rebuttal, most of my concerns are addressed. Hence, I decide to keep my original score and am leaning to accept this paper.

---

### Official Review · Reviewer_zBVq · 2024-07-12

**Soundness:** 3
**Presentation:** 3
**Contribution:** 3
**Rating:** 6
**Confidence:** 3

**Summary:**

This paper proposes a method for generating 3D assets with detailed geometry through inputs from a tactile sensor. More specifically, given a bump-map as input from a tactile sensor (just a small patch is enough), the method uses it as regularization while maximizing the likelihood using a normals-conditioned stable diffusion model. The albedo is also optimized alongside the normals, though limited results are shown. Additionally, using diffusion based segmentation maps, different parts of the image can be given different textures. While each component of the method itself is not too novel, the sum total is and the results are pretty good.

**Strengths:**

1) This is a novel and unexplored task and the proposed method provides the community with a good baseline to build upon.

2) The results of the method are quite convincing. The ability to edit only certain parts of the image with desired textures is particularly nice.

3) The paper is well written with each component explained pretty well.

**Weaknesses:**

1) There are no results on the albedo provided, it would be great if the authors could explain why they were omitted. I strongly urge them to include it in the rebuttal.

2) I may be mistaken, but it seems this method works only for repetitive textures (though the diffusion model is able to change it through optimization). It would be great if the authors could provide the results of an experiment where the object has two very different textures but those textures are only learnt through the diffusion prior. For example, in the cactus pot case, this would correspond to optimizing L_{tactile} only for the pot and let the diffusion prior decide what the normals for the cactus must look like (or vice versa).

**Questions:**

Just reiterating what I already mentioned in the weaknesses: It would be great if the authors could provide the results of an experiment where the object has two very different textures but those textures are only learnt through the diffusion prior. For example, in the cactus pot case, this would correspond to optimizing L_{tactile} only for the pot and let the diffusion prior decide what the normals for the cactus must look like (or vice versa).

I believe this experiment would give us insights on how the diffusion prior itself would perform if a part of the image already has detailed normals (via L_{tactile})

**Limitations:**

Yes

---

> ### Author Rebuttal · Authors · 2024-08-07
>
> Thank you for your encouraging and insightful comments. We’re pleased that you recognize our setup as “a novel and unexplored task.” Below, we address the individual comments.
>
> ### **Add albedo rendering results**
> Due to space limit, we omitted the albedo renderings in the main paper. Thanks for pointing it out and we agree that adding albedo renderings helps to demonstrate the quality more clearly. In the Figure 1 of the attached pdf, we show two examples and will add full results in the revision. Notably, our albedo rendering looks natural and contains minimal baked-in lighting effects or geometric details.
>
> ### **Show experiments of textures learned from the diffusion priors, i.e., optimizing $L_{tactile}$ only for the pot and let the diffusion prior decide what the normals for the cactus must look like (or vice versa)**
> Thanks for your insightful suggestions. We add our experiment results in attached pdf’s Figure 2  for the example “cactus in the pot”, where we keep the text prompt and tactile input unchanged Figure 2(a) has no $L_{tactile}$ for either part; Figure 2(b) and (c) have $L_{tactile}$ for one part, “pot” and “cactus” respectively; Figure 2(d) has $L_{tactile}$ for both parts.
>
> As shown in Figure 2, (b) and (c) contain clear texture for the part with $L_{tactile}$ and synthesize more details for the other parts compared to (a). However, without clear reference, the inferred texture lacks detailed patterns compared to the full results in (d). In short, the texture generated using only diffusion priors appears plausible but tends to look flatter and less detailed.

---

> > ### Comment · Reviewer_zBVq · 2024-08-13
> >
> > I would like to thank the authors for the rebuttal, my concerns were well addressed. Looking at other reviews and the rebuttal, I have decided to raise my score.

---

### Author Rebuttal · Authors · 2024-08-07

We thank all reviewers for their efforts and feedback. The reviewers note that we solve “a novel and unexplored task” (zBVq) with an “innovative approach in utilizing tactile normal modality to enhance geometric details” (MUQX), provide “convincing” (zBVq) and “satisfying” (a9oZ) results, release a TouchTexture dataset that is “beneficial to the research community”  (MUQX, ecYi), and deliver a “structured, well-written” (a9oZ) and “easy to understand” (ecYi) exposition.

The reviewers have suggested additional experiments that will better highlight the strengths and capabilities of our method. **We are happy to report that we have conducted most of the experiments, with favorable results, and we will include them in the revision.** Please see the attached PDF for visual results. Here, we first summarize the new experiments, then provide a more detailed analysis and address other comments and questions in separate threads.

* **Visualizations of albedo renderings (Figure 1)**: As suggested by Reviewer zBVq, we visualize albedo renderings together with the normal and the full-color renderings to present our results. Notably, our albedo rendering looks natural and contains minimal baked-in lighting effects or geometric details.
* **Ablation study of tactile matching loss $L_{tactile}$ on different parts of an object (Figure 2)**: Following Reviewer zBVq’s advice, we show the “cactus in the pot” example where only a part of the object has $L_{tactile}$. The texture learnt purely from diffusion prior tends to be flatter and less detailed compared to those learned with  $L_{tactile}$.
* **Generating mesh using real image and corresponding tactile signal (Figure 3)**: As suggested by Reviewer MUQX and a9oZ, we generate a mesh from real data input, and our method achieves reasonable reconstruction with color and geometric details.
* **Integrating our method with more recent baselines (Figure 4 and Figure 5)**: As suggested by Reviewer MUQX and ecYi, We integrate RichDreamer (Figure 4), a purely text-to-3D baseline, and InstantMesh (Figure 5), a recent image-to-3D baseline into our method, demonstrating our method’s compatibility with various pipelines. Our method, when integrated with these baselines, generalizes well to complex objects across different genres, and achieves better results compared to using the baselines alone.
* **Quantitative Evaluation of our diffusion-based segmentation method**: To answer Reviewer MUQX’s question, we quantitatively evaluate our segmentation method by manually annotating a test set of meshes. Our method achieves an accuracy of 83.7%, demonstrating its capability to provide sufficient segmentation for multi-part optimization.

---

### Decision · Program_Chairs · 2024-09-25

**Decision:**

Accept (poster)

**Comment:**

This submission has garnered 1 accept, 2 weak accepts, and 1 borderline reject. Reviewers zBVq, MUQX, and a9oZ support the paper, recognizing its novelty and the convincing nature of its contributions. The authors have successfully addressed the concerns raised during the review process, which include issues of repetitive textures, adaptability to complex objects, and the role of tactile sensing. Reviewer ecYi expressed reservations, comparing the proposed text-driven 3D generation method to traditional 3D reconstruction approaches; however, the authors' rebuttal effectively clarified the distinctions between these methods. Given the supportive majority among the reviewers, the AC recommends acceptance of this manuscript.